# INFERENCE FROM REAL-WORLD SPARSE MEASUREMENTS

## ABSTRACT

Real-world problems often involve complex and unstructured sets of measurements, which occurs when sensors are sparsely placed in either space or time. Being able to model this irregular spatiotemporal data and extract meaningful forecasts is crucial. Deep learning architectures capable of processing sets of measurements with positions varying from set to set, and extracting readouts anywhere are methodologically difficult. Current state-of-the-art models are graph neural networks and require domain-specific knowledge for proper setup.

We propose an attention-based model focused on robustness and practical applicability, with two key design contributions. First, we adopt a ViT-like transformer that takes both context points and read-out positions as inputs, eliminating the need for an encoder-decoder structure. Second, we use a unified method for encoding both context and read-out positions. This approach is intentionally straightforward and integrates well with other systems. Compared to existing approaches, our model is simpler, requires less specialized knowledge, and does not suffer from a problematic bottleneck effect, all of which contribute to superior performance.

We conduct in-depth ablation studies that characterize this problematic bottleneck in the latent representations of alternative models that inhibit information utilization and impede training efficiency. We also perform experiments across various problem domains, including high-altitude wind nowcasting, two-day weather forecasting, fluid dynamics, and heat diffusion. Our attention-based model consistently outperforms state-of-the-art models in handling irregularly sampled data. Notably, our model reduces the root mean square error (RMSE) for wind nowcasting from 9.24 to 7.98 and for heat diffusion tasks from 0.126 to 0.084.

## 1 INTRODUCTION

Deep learning (DL) has emerged as a powerful tool for modeling dynamical systems by leveraging vast amounts of data available in ways that traditional solvers cannot. This has led to a growing reliance on DL models in weather forecasting, with state-of-the-art results in precipitation nowcasting Suman et al. (2021); Shi et al. (2017) and performance on par with traditional partial differential equation (PDE) solvers in medium-term forecasting Lam et al. (2022). However, these applications are currently limited to data represented as images or on regular grids, where models such as convolutional networks or graph neural networks are used. In contrast, various real-world data often comes from irregularly placed or moving sensors, which means custom architectures are needed to handle it effectively.

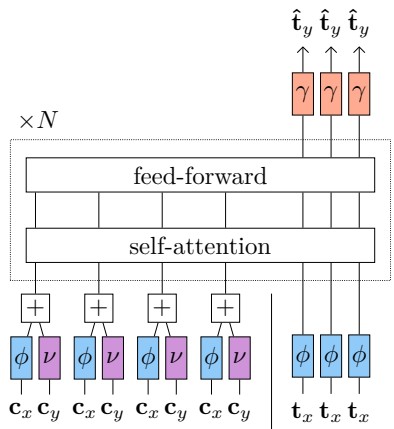

Figure 1: Multi-layer Self-Attention

An example which can benefit significantly from such an architecture is Air Traffic Control (ATC). ATC needs reliable weather forecasts to manage airspace efficiently. This is particularly true for wind conditions, as planes are highly sensitive to wind and deviations from

| Architecture | Performance | Simplicity | Domain Knowledge Agnostic |
|---|---|---|---|
| CNP | ✗ | ✓✓ | ✓✓ |
| GEN | ✓ | ✗ | ✗ |
| TFS | ✓ | ✓ | ✓✓ |
| MSA (Ours) | ✓✓ | ✓✓ | ✓✓ |

Table 1: Comparison between our approach and the different baselines. Multi-Layer Self-Attention (MSA) achieves good performance while being simple to implement and does not require practitioner knowledge for proper setup.

the initial flight plan can be costly and pose safety hazards. DL models are a promising candidate for producing reliable wind forecasts as a large amount of data is collected from airplanes that broadcast wind speed measurements with a four seconds frequency. A model that can effectively model wind speeds using data collected from airplanes,should be able to decode anywhere in space, as we aim to predict wind conditions at future locations of the airplane, conditioned on past measurements taken by that specific airplane or neighboring ones in a permutation invariant manner.

To meet these requirements, we introduce a Multi-layer Self-Attention model (MSA) and compare it to different baselines [Table 1]: Conditional Neural Processes (CNP) Garnelo et al. (2018), Graph Element Networks (GEN) Alet et al. (2019) and a transformer encoder-decoder baseline (TFS) that we developed. While all of these models possess the aforementioned characteristics, they each adopt distinct strategies for representing measurements within the latent space. CNP models use a single vector as a summary of encoded measures, while GEN models map the context to a graph, based on their distance to its nodes. MSA keeps one latent vector per encoded measurement and can access them directly for forecasting. This latent representation is better, as it does not create a bottleneck. We show that due to that architectural choice, both baselines can fail, in certain cases, to retrieve information present in the context they condition on.

Our approach offers better performance than its competitors and is conceptually simpler as it does not require an encoder-decoder structure. To evaluate the effectiveness of our approach, we conducted experiments on high-altitude wind nowcasting, heat diffusion and fluid dynamics and two-day weather forecasting. Several additional ablation studies show the impact of different architectural choices.

The main contributions of this work are summarized below:

- We develop an attention-based model that can generate prediction anywhere in the space conditioned on a set of measurements.

- We propose a novel encoding scheme using a shared MLP encoder to map context and target positions, improving forecasting performance and enhancing the model's understanding of spatial patterns.

- We evaluate our method on a set of challenging tasks with data irregularly sampled in space: high-altitude wind nowcasting, two-day weather forecasting, heat diffusion and fluid dynamics.

- We examine the differences between models, and explain the impact of design choices such as latent representation bottlenecks on the final performance of the trained models.

## 2 RELATED WORKS

DL performance for weather forecasting has improved in recent years, with DL models increasingly matching or surpassing the performance of traditional PDE-based systems. Initially applied to precipitation nowcasting based on 2D radar images Suman et al. (2021); Shi et al. (2017), DL-based models have recently surpassed traditional methods for longer forecast periods Lam et al. (2022). In the case of radar precipitation, data is organized as images and convolutional neural networks are utilized. For 3D regular spherical grid data, graph neural networks or spherical CNNs are employed Lam et al. (2022); Esteves et al. (2023). However, in our study, the data set is distributed sparsely in space, which hinders the use of these traditional architectures. The use of DL for

modelling dynamical systems, in general, has also seen recent advancements Li et al. (2021); Gupta & Brandstetter (2022); Pfaff et al. (2020) but most approaches in this field typically operate on regularly-spaced data or on irregular but fixed mesh.

Neural Processes Garnelo et al. (2018); Kim et al. (2019), Graph Element Networks Alet et al. (2019) and attention-based models Vaswani et al. (2017) are three DL-based approaches that are capable of modeling sets of data changing from set to set. In this study, we conduct a comparison of these models by selecting a representative architecture from each category. Additionally, attention-based models have been previously adapted for set classification tasks Lee et al. (2019), and here we adapt them to generate forecasts.

Pannatier et al. (2021) use a kernel-based method for wind nowcasting based on flight data. This method incorporates a distance metric with learned parameters to combine contexts for prediction at any spatial location. However, a notable limitation of this technique is that its forecasts are constrained to the convex hull of the input measurements, preventing accurate extrapolation. We evaluate the efficacy of our method compared to this approach, along with the distinct outcomes obtained, in Section 5 of the supplementary material.

While previous studies have utilized transformers for modeling physical systems Geneva & Zabaras (2022), time series Li et al. (2019) or trajectory predictions Girgis et al. (2022); Nayakanti et al. (2023); Yuan et al. (2021) these applications do not fully capture the specific structure of our particular domain, which involves relating two spatial processes at arbitrary points on a shared domain. Although we model temporal relationships, our approach lacks specialized treatment of time. Therefore, it does not support inherently time-based concepts like heteroskedasticity, time-series imputation, recurrence, or seasonality. Further details distinguishing our approach from other transformer-based applications are elaborated in Section 3 of the supplementary material.

## 3 METHODOLOGY

### 3.1 CONTEXT AND TARGETS

The problem addressed in this paper is the prediction of target values given a context and a prediction target position. Data is in the form of pairs of vectors $(\mathbf{c}_x, \mathbf{c}_y)$ and $(\mathbf{t}_x, \mathbf{t}_y)$ where $\mathbf{c}_x$ and $\mathbf{t}_x$ are the position and $\mathbf{c}_y$ and $\mathbf{t}_y$ are the measurements (or values), where we use $c$ for context, $t$ for target, $x$ for spatial position and $y$ for the corresponding vector value. The positions lie in the same underlying space $\mathbf{c}_x, \mathbf{t}_x \in \mathbb{X} \subseteq \mathbb{R}^X$, but the context and target values not necessarily. We define the corresponding spaces as $\mathbf{c}_y \in \mathbb{I} \subseteq \mathbb{R}^I$ and $\mathbf{t}_y \in \mathbb{O} \subseteq \mathbb{R}^O$, respectively, where $X, I, O$ are integers that need not be equal. The data set consists of multiple pair of context and target sets that can be of different lengths, we denote the length of the $j$-st context, target set respectively $N_c^j$ and $N_t^j$.

All models take as input a set of context pairs $\{(\mathbf{c}_x, \mathbf{c}_y)_i^j\}_{i=1}^{N_c^j}$, as well as target positions, denoted $\{(\mathbf{t}_x)_i^j\}_{i=1}^{N_t^j}$.

As an example, to transform a data set of wind speed measurements into context and target pair, we partitioned the data set into one-minute time segments and generated context and target sets with an intervening delay, as depicted in Figure 2. The underlying space, denoted by $\mathbb{X}$, corresponds to 3D Euclidean space, with both $\mathbb{I}$ and $\mathbb{O}$ representing wind speed measurements in the $x, y$ plane. The models are given a set of context points at positions $c_x$ of value $c_y$, and should be able when given target positions $t_x$ to output a corresponding value $t_y$ conditioned on the context. Detailed descriptions of the data set, including illustration of the different problems, and the respective spaces for other scenarios and the ablation study can be found in Table 2 within the supplementary material.

### 3.2 ENCODING SCHEME

We propose in this section a novel encoding scheme for irregularly sampled data. Our approach leverages the fact that both the context measurements and target positions reside within a shared underlying space. To exploit this shared structure, we adopt a unified two-layers MLP encoder $\phi$ for mapping both the context and target position to a latent space representation. Then, we use a second MLP $\nu$ to encode the context values and add them to the encoded positions when available. This differs from the approach proposed in Garnelo et al. (2018); Alet et al. (2019) where both the context

position and value are concatenated and given to a encoder, and the target position is encoded by another. The schemes are contrasted as:

$$\mathbf{c}_e = \varphi(\mathbf{c}_x, \mathbf{c}_y) \qquad (1) \qquad\qquad \mathbf{c}_e = \phi(\mathbf{c}_x) + \nu(\mathbf{c}_y) \qquad (3)$$

$$\mathbf{t}_e = \psi(\mathbf{t}_x) \qquad (2) \qquad\qquad \mathbf{t}_e = \phi(\mathbf{t}_x) \qquad (4)$$

Traditional methods            Proposed scheme

Where $\phi, \nu, \varphi, \psi$ are two hidden-layers MLPs, and $\mathbf{c}_e, \mathbf{t}_e \in \mathbb{R}^E$ are the encoded measurements positions and values and encoded target position respectively.

## 3.3 MULTI-LAYER SELF-ATTENTION (MSA, OURS)

Our proposed model, Multi-layer Self-Attention (MSA) harnesses the advantages of attention-based models. MSA maintains a single latent representation per input measurement and target position, which conveys the ability to propagate gradients easily and correct errors in training quickly. MSA can access and combine target position and context measurements at the same time, which forms a flexible and powerful method for approaching the latent space. Our model is similar to a transformer-encoder, as the backbone of a ViT Dosovitskiy et al. (2020), it can be written as:

$$\mathbf{c}_l, \mathbf{t}_l = \text{Transformer-Encoder}(\mathbf{c}_e, \mathbf{t}_e) \qquad \mathbf{c}_e \in \mathbb{R}^{N_C \times E}, \mathbf{t}_e \in \mathbb{R}^{N_t \times E} \qquad (5)$$

$$\hat{\mathbf{t}}_y = \gamma(\mathbf{t}_l) \qquad\qquad \hat{\mathbf{t}}_y \in \mathbb{R}^{\mathbb{N}_t \times O} \qquad (6)$$

MSA does not use positional encoding for encoding the order of the inputs. This model is permutation equivariant due to the self-attention mechanism and it uses full attention, allowing each target feature to attend to all other targets and context measurements. MSA generate all the output in one pass in a non-autoregressive way and the outputs of the model are only the units that correspond to the target positions, which are then used to compute the loss.

## 3.4 BASELINES

**Transformer(s) (TFS)** We also adapt an encoder-decoder transformer (TFS) model Vaswani et al. (2017). The motivation behind this was the intuitive appeal of the encoder-decoder stack for this specific problem. TFS in our approach deviates from the standard transformer in a few ways: Firstly, it does not employ causal masking in the decoder and secondly, the model forgoes the use of positional encoding for the sequence positions. It can be written as:

$$\mathbf{c}_l = \text{Transformer-Encoder}(\mathbf{c}_e) \qquad \mathbf{c}_e \in \mathbb{R}^{N_t \times E} \qquad (7)$$

$$\mathbf{t}_l = \text{Transformer-Decoder}(\mathbf{c}_l, \mathbf{t}_e) \qquad \mathbf{c}_l, \mathbf{t}_e \in \mathbb{R}^{N_t \times E} \qquad (8)$$

$$\hat{\mathbf{t}}_y = \gamma(\mathbf{t}_l) \qquad \hat{\mathbf{t}}_y \in \mathbb{R}^{\mathbb{N}_t \times O} \qquad (9)$$

In comparison with MSA, TFS uses an encoder-decoder architecture, which adds a layer of complexity. Moreover, it necessitates the propagation of error through two pathways, specifically through a cross-attention mechanism that lacks a residual connection to the encoder inputs.

**Graph Element Network(s) (GEN)** Graph Element Networks (GEN) Alet et al. (2019) is an architecture that utilizes a graph $\mathcal{G}$ as a latent representation. The encoder maps measurements to the nodes of the graph based on their distance to the nodes, and the nodes' features are processed by $L$ iteration of message passing. The nodes positions and edges of the graphs are additional parameters that must be carefully chosen. Additionally the nodes position can be optimized during training. The whole model can be described as:

$$\mathbf{n}_e = \sum_e r(\mathbf{c}_x, \mathbf{n}_\mathbf{x})\mathbf{c}_e \qquad \mathbf{c}_e \in \mathbb{R}^{N_t \times E} \qquad (10)$$

$$\mathbf{n}_l = \text{Message-Passing}(\mathbf{n}_e, \mathcal{G}, L) \qquad \mathbf{n}_e \in \mathbb{R}^{N_n \times E} \qquad (11)$$

$$\mathbf{c}_l = \sum_l r(\mathbf{n}_\mathbf{x}, \mathbf{t}_x)\mathbf{n}_l \qquad \mathbf{n}_l \in \mathbb{R}^{N_n \times E} \qquad (12)$$

$$\hat{\mathbf{t}}_y = \gamma(\mathbf{c}_l, \mathbf{t}_l) \qquad \hat{\mathbf{t}}_y \in \mathbb{R}^{\mathbb{N}_t \times O} \qquad (13)$$

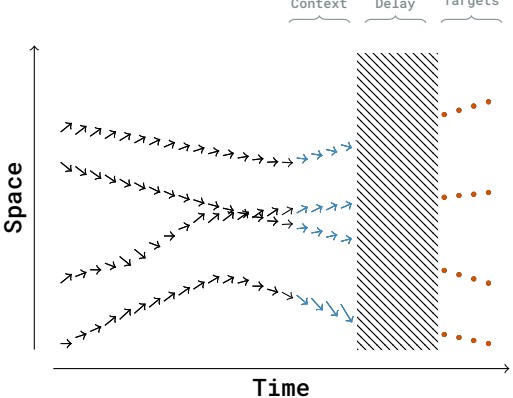

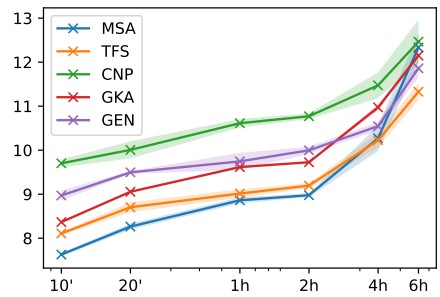

Figure 2: Description of the context and target sets in the wind nowcasting case. The context set and the target set are time slices separated by a delay, which corresponds to the forecasting window. The underlying space is in that case $\mathbb{X} \subseteq \mathbb{R}^3$ and the context values and target values both represent wind speed and belong to the same space $\mathbb{I} = \mathbb{O} \subseteq \mathbb{R}^2$.

Figure 3: RMSE of the different models depending on the forecast duration (lower is better). We ran three experiments varying the pseudorandom number generator seeds for each time window and each model to measure the standard deviation. The error does not increase drastically over the first two hours because the wind has some persistence and the context values are good predictors of the targets in that regime.

GEN has for inductive bias that a single latent vector summarize a small part of the space. As it includes a distance-based encoding and decoding scheme, the only way for the model to learn non-local patterns is through message passing. This model was originally designed with a simple message-passing scheme. But it can easily be extended to a broad family of graph networks by using different message-passing schemes, including ones with attention. We present some related experiments in Section 9 of the supplementary material.

**Conditional Neural Process(es) (CNP)**  CNP Garnelo et al. (2018) encodes the whole context as a single latent vector. They can be seen as a subset of GEN. Specifically, a CNP is a GEN with a graph with a single node and no message passing. While CNP possess the desirable property of being able to model any permutation-invariant function Zaheer et al. (2017), their expressive capability is constrained by the single node architecture Kim et al. (2019). Despite this, CNP serve as a valuable baseline and are considerably less computationally intensive.

$$\mathbf{c}_l = \text{mean}(\mathbf{c}_e) \qquad\qquad \mathbf{c}_e \in \mathbb{R}^{N_C \times E} \qquad\qquad (14)$$

$$\hat{\mathbf{t}}_y = \gamma(\mathbf{t}_l, \mathbf{c}_l) \qquad\qquad \hat{\mathbf{t}}_y \in \mathbb{R}^{\mathbb{N}_t \times O} \qquad\qquad (15)$$

## 4  EXPERIMENTS

Our experiments aim to benchmark the performance of our models on various data sets with irregularly sampled data. The first task focuses on high-altitude wind nowcasting. The second task is on heat diffusion. Additionally, we evaluate our models on fluid flows, considering both a steady-state case governed by the Darcy Flow equation and a dynamic case modeling the Navier-Stokes equation in an irregularly spaced setting. Finally, we compare the models on a weather forecasting task, utilizing irregularly sampled measurements from the ERA5 data set Hersbach et al. (2023) to predict wind conditions two days ahead.

For the Wind Nowcasting Experiment, the data set, described in Section 3.1, consists of wind speed measurements collected by airplanes with a sampling frequency of four seconds. We evaluate our models on this data set [Table 2] and we assess the models' performance as a function of forecast duration, as depicted in Figure 3. We select model configurations with approximately 100,000 parameters and run each model using three different random seeds. Our results indicate that attention-based models consistently outperform other models for most forecast durations, except for in the

Table 2: Validation RMSE of the High-Altitude Wind Nowcasting, Poisson, Navier Stokes and Darcy Flow equation and the weather forecasting task. Each model ran for 10, 2000, 1000, 100 and 100 epochs respectively on an NVIDIA GeForce GTX 1080 Ti. The low number of epochs for wind nowcasting is due to the amount of data which is considerably larger than in the other experiments. The standard deviation is computed over 3 runs. We present here the original implementation of CNP and GEN compared with TFS and MSA with sharing weights for the position. More details can be found in Table 1 of the supplementary material. We choose the configuration of the models so that every model has a comparable number of parameters. We underline the best models for each size and indicate in bold the best model overall.

| Architecture | Size | Wind Nowcasting | Poisson Equation | Navier Stokes Equation | Darcy Flow Equation | ERA5 |
|---|---|---|---|---|---|---|
| **CNP** | 5k | $11.94_{\pm\,0.78}$ | $0.33_{\pm\,0.004}$ | $0.701_{\pm\,0.0023}$ | $0.0311_{\pm\,0.0008}$ | $2.129_{\pm\,0.0039}$ |
| | 20k | $10.19_{\pm\,1.83}$ | $0.32_{\pm\,0.003}$ | $0.672_{\pm\,0.0011}$ | $0.0295_{\pm\,0.0002}$ | $2.117_{\pm\,0.0018}$ |
| | 100k | $10.17_{\pm\,1.24}$ | $0.33_{\pm\,0.003}$ | $0.656_{\pm\,0.0007}$ | $0.0286_{\pm\,0.0001}$ | $2.110_{\pm\,0.0002}$ |
| **GEN** | 5k | $11.02_{\pm\,3.19}$ | $0.12_{\pm\,0.006}$ | $0.604_{\pm\,0.0010}$ | $0.0304_{\pm\,0.0003}$ | $2.132_{\pm\,0.0035}$ |
| | 20k | $9.98_{\pm\,0.76}$ | $0.13_{\pm\,0.014}$ | $0.599_{\pm\,0.0006}$ | $0.0296_{\pm\,0.0002}$ | $2.124_{\pm\,0.0031}$ |
| | 100k | $9.56_{\pm\,0.21}$ | $0.16_{\pm\,0.049}$ | $0.596_{\pm\,0.0005}$ | $0.0294_{\pm\,0.0001}$ | $2.121_{\pm\,0.0005}$ |
| **TFS (Ours, baseline)** | 5k | $8.30_{\pm\,0.03}$ | $0.15_{\pm\,0.036}$ | $0.604_{\pm\,0.0022}$ | $0.0275_{\pm\,0.0014}$ | $2.129_{\pm\,0.0032}$ |
| | 20k | $8.20_{\pm\,0.04}$ | $0.09_{\pm\,0.006}$ | $0.596_{\pm\,0.0008}$ | $\underline{0.0258}_{\pm\,\mathbf{0.0003}}$ | $2.109_{\pm\,0.0012}$ |
| | 100k | $8.38_{\pm\,0.13}$ | $0.18_{\pm\,0.014}$ | $0.591_{\pm\,0.0012}$ | $0.0269_{\pm\,0.0004}$ | $2.100_{\pm\,0.0011}$ |
| **MSA (Ours)** | 5k | $8.07_{\pm\,0.11}$ | $0.11_{\pm\,0.006}$ | $0.597_{\pm\,0.0011}$ | $0.0274_{\pm\,0.0011}$ | $2.125_{\pm\,0.0070}$ |
| | 20k | $\underline{7.98}_{\pm\,\mathbf{0.03}}$ | $\mathbf{0.08}_{\pm\,\mathbf{0.003}}$ | $\underline{0.589}_{\pm\,0.0013}$ | $0.0259_{\pm\,0.0007}$ | $2.107_{\pm\,0.0020}$ |
| | 100k | $\underline{8.18}_{\pm\,0.14}$ | $\underline{0.10}_{\pm\,0.009}$ | $\mathbf{0.589}_{\pm\,\mathbf{0.0006}}$ | $0.0264_{\pm\,0.0004}$ | $\mathbf{2.098}_{\pm\,\mathbf{0.0029}}$ |

Table 3: Evaluation of the wind nowcasting task according to standard weather metrics, which are described in Section 6 of the supplementary material. The optimal value of the metric is indicated in the parenthesis. MSA is the best model overall, with the lowest absolute error, a near-zero systematical bias and output values that have a similar dispersion to GEN.

| Model | RMSE ($\downarrow$) | $\theta$ MAE ($\downarrow$) | $r$ MAE ($\downarrow$) | Relative BIAS$_x$ (0.0) | Relative BIAS$_y$ (0.0) | rSTD (1.0) | NSE ($\uparrow$) |
|---|---|---|---|---|---|---|---|
| **CNP** | $10.99_{\pm\,0.75}$ | $25.55_{\pm\,1.22}$ | $9.22_{\pm\,0.33}$ | $0.00_{\pm\,0.09}$ | $-1.09_{\pm\,0.03}$ | $1.25_{\pm\,0.07}$ | $-0.23_{\pm\,0.01}$ |
| **GEN** | $8.97_{\pm\,0.06}$ | $22.56_{\pm\,0.77}$ | $6.97_{\pm\,0.05}$ | $-0.02_{\pm\,0.03}$ | $-0.97_{\pm\,0.21}$ | $\mathbf{1.09}_{\pm\,\mathbf{0.07}}$ | $0.25_{\pm\,0.02}$ |
| **GKA** | $8.44_{\pm\,0.01}$ | $21.89_{\pm\,0.02}$ | $6.65_{\pm\,0.02}$ | $-0.02_{\pm\,0.00}$ | $-1.78_{\pm\,0.02}$ | $1.13_{\pm\,0.00}$ | $0.31_{\pm\,0.01}$ |
| **TFS (Ours, baseline)** | $7.99_{\pm\,0.15}$ | $22.17_{\pm\,1.20}$ | $6.48_{\pm\,0.50}$ | $0.08_{\pm\,0.10}$ | $-2.21_{\pm\,2.67}$ | $1.17_{\pm\,0.04}$ | $0.43_{\pm\,0.08}$ |
| **MSA (Ours)** | $\mathbf{7.36}_{\pm\,\mathbf{0.06}}$ | $\mathbf{20.48}_{\pm\,\mathbf{0.48}}$ | $\mathbf{5.67}_{\pm\,\mathbf{0.11}}$ | $\mathbf{0.00}_{\pm\,\mathbf{0.02}}$ | $\mathbf{-0.04}_{\pm\,\mathbf{0.64}}$ | $\mathbf{1.09}_{\pm\,\mathbf{0.02}}$ | $\mathbf{0.55}_{\pm\,\mathbf{0.05}}$ |

6-hour range. Notably, we found that the Gaussian Kernel Averaging (GKA) model used in previous work Pannatier et al. (2021) achieves satisfactory performance, despite its theoretical limitations, which we analyse in Section 5 of the supplementary material. Moreover, our findings suggest that attention-based models, particularly MSA and TFS, exhibit superior performance in this setup. Furthermore, we observe that the GKA model performed well for short time horizons when most of the information in the context was still up-to-date. However, as the time horizon increased, the GKA model's lack of flexibility become more apparent, and GEN become more competitive.

For the Heat Diffusion Experiment, we utilize the data set introduced in Alet et al. (2019), derived from a Poisson Equation solver. The data set consists of context measurements in the unit square corresponding to sink or source points, as well as points on the boundaries. The targets correspond to irregularly sampled heat measurements in the unit cube. Our approach offers significant performance improvements, reducing the root mean square error (RMSE) from 0.12 to 0.08, (MSE reduction of 0.016 to 0.007, in terms of the original metric) as measured against the ground truth [Table 2].

For the Fluid Flow Experiment, both data sets are derived from Li et al. (2021), subsampled irregularly in space. In both cases, our models outperforms the alternative [Table 2]. In the Darcy Flow equation, the TFS model with 20k parameters exhibits the best performance, but this task proved to be relatively easier, and we hypothesize that the MSA model could not fully exploit this specific setup. However, it is worth mentioning that the performance of the MSA model was within a standard deviation of the TFS model.

We conducted a Two-Day Weather Forecasting Experiment utilizing ERA5 data set measurements. The data set consists of irregularly sampled measurements of seven quantities, including wind speed at different altitudes, heat, and cloud cover. Our goal is to predict wind conditions at 100 meters two days ahead based on these measurements. MSA demonstrates its effectiveness in capturing the temporal and spatial patterns of weather conditions, enabling accurate predictions [Table 2].

To summarize, our experiments encompass a range of tasks including high-altitude wind nowcasting, heat diffusion, fluid modeling, and two-day weather forecasting. Across these diverse tasks and data sets, our proposed model consistently outperforms baseline models, showcasing their efficacy in capturing complex temporal and spatial patterns.

## 5 UNDERSTANDING FAILURE MODES

We examine the limitations of CNP and GEN latent representation for encoding a context. Specifically, we focus on the bottleneck effect that arises in CNP from using a single vector to encode the entire context, resulting in an underfitting problem Garnelo et al. (2018); Kim et al. (2019), and that applies similarly to GEN. To highlight this issue, we propose three simple experiments. (1) We show in which case baselines are not able to retrieve information in the context that they use for conditioning, and why MSA and TFS are not suffering from this problem. (2) We show that maintaining disentangled latent representation helped to the correct attribution of perturbations. (3) We show that this improved latent representation leads to better error correction.

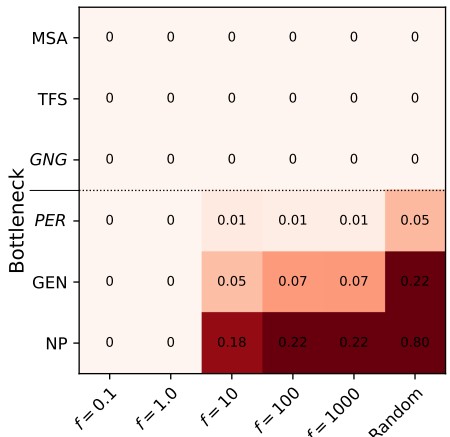

Figure 4: Results of the information retrieval experiment. The three first rows correspond to models with no bottlenecks. The x-axis corresponds to data sets created by increasing frequency. Random is the extreme case were the context value is independent from the position. When the learned function does not vary too much spatially, models with bottlenecks can suffice. The models in italics represent hybrid architectures: *GNG* = GEN without a graph, which maintains a latent per context measure and *PER* = transformer with a perceiver layer Jaegle et al. (2021) which creates a bottleneck.

### 5.1 CONTEXT INFORMATION RETRIEVAL

Every model considered in this work encodes context information differently. Ideally, each should be capable of using or retrieving every measure in their context. We will see that excessive bottlenecking in the latent space can make this difficult or impossible.

To demonstrate this result, we design a simple experiment in which each model encodes a set of $64$ measures $(\mathbf{c}_x, \mathbf{c}_y)$, and is then tasked with retrieving the corresponding $\mathbf{t}_y = \mathbf{c}_y$ given the $\mathbf{t}_x = \mathbf{c}_x$. The training and validation set have respectively $10\,000$ and $1\,000$ pairs of sets of $64$ examples. It is worth noting that the models have access to all the information they need to solve the task with near-zero MSE. We conducted several experiments, starting by randomly sampling 2D context positions $\mathbf{c}_x = (x, y)$ from a Gaussian distribution and computing the associated deterministic smooth function:

$$\mathbf{c}_y = \sin(\pi f x) \cos(\pi f y) \in \mathbb{R} \tag{16}$$

where $f$ is a frequency parameter that governs problem difficulty. The higher $f$ is, the more difficult the function becomes, as local information becomes less informative about the output. We also consider as a harder problem to sample $c_y$ randomly and independently from the position.

The results of this experiment, as shown in Figure 4, indicate that the CNP and GEN models are less effective in learning this task at higher frequencies. This inefficiency is primarily due to a phenomenon we define as a 'bottleneck': a situation where a single latent variable is responsible for representing two distinct context measurements. This bottleneck impedes the models' ability to distinguish and retrieve the correct target value. In contrast, models with disentangled latent representations, like

MSA, are not subject to this limitation and thus demonstrate superior performance in learning the task.

To further demonstrate this bottleneck effect, we created two hybrid models. The first one denoted GNG (for *GEN No Graph*), is adapted from GEN but instead of relying on a common graph, creates one based on the measure position with one node per measure. Edges are artificially added between neighboring measures which serve as the base structure for $L$ steps of message-passing. This latent representation is computationally expensive as it requires the creation of a graph per set of measurements, but it does not create a bottleneck in the latent representation. We found that GNG is indeed able to learn the task at hand. We then followed the reverse approach and artificially added a bottleneck in the latent representation of attention-based models by using Perceiver Layer Jaegle et al. (2021) with $P$ learned latent vectors instead of the standard self-attention in the transformer encoder (and call the resultant model *PER*). When $P$ is smaller than the number of context measurements, it creates a bottleneck and *PER* does not succeed in learning the task. If the underlying space is smooth enough, GEN, CNP and *PER* are capable of reaching perfect accuracy on this task as they can rely on neighboring values to retrieve the correct information. This experiment demonstrates that MSA and TFS can use their disentangled latent representations to efficiently retrieve context information regardless of the level of discontinuity in the underlying space, while models with bottlenecks, such as CNP and GEN, are limited in this regard and perform better when the underlying space is smooth.

## 5.2 CORRECT PERTUBATION ATTRIBUTION

In the following analysis, we explore how disentangled latent representations can enhance error correction during training. Specifically, we ask whether models can correctly attribute the effects of a perturbation in the output to backpropagation.

We use MSA, TFS and GEN models applied to the information retrieval task discussed in the previous section. We pre-train each model to zero error on the validation data in a smooth case ($f = 1.0$), then apply a perturbation $\gamma$ on one of the output values, compute the loss between the perturbated output and the one without perturbation and backpropagate it through each model. The norm of gradients corresponding to the latent at the last encoder layer is shown in Figure 5. We see that MSA and TFS only receive a signal on the corresponding latent while the other models receive signals on different latents. Here MSA has 128 latent vectors as it maintains one latent per context measurement and per target position, and we see that the model has a signal only on the latent corresponding to the position where there is an error.

We hypothesize that this interference of gradients to other non-related latents impedes training, as the models struggle to correct the artificial error while maintaining the same value for

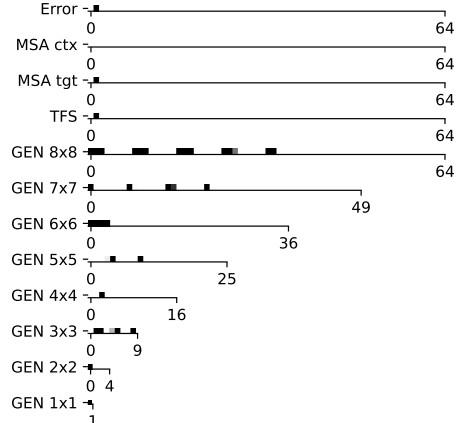

Figure 5: Gradients on the last layer of the encoder corresponding to an artificial error of $\gamma = 10.0$ added to the second output. MSA maintains a disentangled representation and the gradient at that layer is non-zero only on the corresponding latent. We compare it to different GEN models each initialized with a graph corresponding to a regular grid of size $i \times i$ with $i \in \{1, \ldots, 8\}$. Due to the bottleneck effect, the gradients corresponding to one error are propagated across different latent vectors for GEN. Even when there are enough latents (GEN $8 \times 8$), GEN still disperse attribution because their distance-based conditioning that does not allow for a one-to-one mapping between targets and latents.

the other output values. Models with a disentangled representation can update the corresponding latent independently and follow a smoother optimization trajectory.

To demonstrate this property, for the models described above, we tabulate the number of backpropagation passes needed to fully correct the artificial error on one output (to reach a zero error on the validation set again).

The results are shown in Figure 6. We used MSA as a reference and observed that all models with an entangled representation required more time to reach a zero error on the validation set. We found that

the more entangled the representation, the more time was needed to reach the desired performance. Note that, in this setup, GEN $1 \times 1$ is equivalent to CNP.

These experiments demonstrate that models with disentangled latent representations can more efficiently correct errors during training, while models with entangled representations struggle to do so and require more time to reach the desired performance.

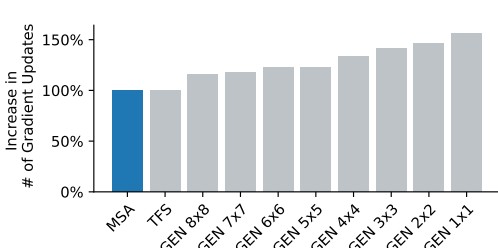

### 5.3 ENCODING SCHEME

In this section, we evaluate the novel encoding scheme presented in Section 3.2, we present the results in Table 1 of the supplementary material. We found that, it reduces the RMSE from $8.47$ to $7.98$ in the wind nowcasting task and enables the MSA model to achieve the best performance with an RMSE of $0.08$ for the Poisson Equation. Sharing the same mapping for positions is the appropriate inductive bias for encoding positions, as it eliminates the need to learn the same transformation twice. Since our data is irregularly sampled in space, the positioning of measurements and target positions significantly influences the prediction, as demonstrated in additional experiments, Sections 7 and 8 in the

Figure 6: Comparison of the number of gradient updates required to correct an artificial error, with respect to MSA (lower is better). The y-axis represents the increase in percentage in the number of steps required to reach a perfect accuracy with respect to MSA. We compared MSA to different GEN each initialized with a graph corresponding to a regular grid of size $i \times i$ with $i \in \{1, \ldots, 8\}$. It can be observed that all GEN take more steps to correct the same mistake, and the more entangled the latent representation is, the more time it requires to correct the problem.

supplementary material. We think that sharing the position mapping can link information from the context and target positions, which helps the model to understand better how the space is shaped.

## 6 CONCLUSION

In this work, we introduced an attention-based model to handle the challenges of wind nowcasting data. We demonstrated that the proposed attention-based model was able to reach the best performance for high-altitude wind prediction and other dynamical systems, such as weather forecasting, heat diffusion and fluid dynamics when working with data irregularly sampled in space. We then explained why attention-based models were capable of outperforming other models on that task and provided an in-depth examination of the differences between models, providing explanations for the impact of design choices such as latent representation bottlenecks on the final performance of the trained models.

Our work builds upon well-established attention models, which have demonstrated their versatility and efficacy in various domains. Although the core model is essentially a vanilla transformer, our architecture required careful adaptation to suit our specific requirements. We designed our model to be set-to-set rather than sequence-to-sequence, handling data in a non-causal and non-autoregressive manner, and generating continuous values for regression. The success of influential models like BERT Devlin et al. (2019), GPT Radford et al. (2018), ViT Dosovitskiy et al. (2020), and Whisper Radford et al. (2022), also closely resemble the original implementation by Vaswani et al. (2017), which further supports the effectiveness of the transformer framework across different tasks and domains.

Finally, our model's scalability is currently limited by its quadratic complexity in the context size. Although this limitation does not pose a problem in our particular use cases, it can impede the scaling of applications. This is a significant challenge that affects all transformer-based models and has garnered considerable attention. Recent developments to tackle this challenge include flash-attention Dao et al. (2022), efficient transformers Katharopoulos et al. (2020), and quantization techniques Dettmers et al. (2022), which can address this problem, enhancing the feasibility of our approach for large-scale applications.

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
