# OpenReview forum: "Inference from Real-World Sparse Measurements"
_ICLR.cc/2024/Conference — ICLR 2024 Conference Withdrawn Submission_

### Official Review · Reviewer_7eM6 · 2023-10-25

**Soundness:** 2 fair
**Presentation:** 2 fair
**Contribution:** 2 fair
**Rating:** 3
**Confidence:** 3

**Summary:**

This paper proposes an attention-based method for handling irregularly sampled data. The proposed method uses ViT-like transformer architecture to combine context points and read-out positions. Compared with other baseline models, the proposed method shows better performance in data forecasting tasks based on irregularly sampled data.

**Strengths:**

- This paper tackles irregularly sampled datasets, which is an under-explored area in the scientific ML community.

- The proposed method is simple and powerful. It is applicable to data from different domains.

**Weaknesses:**

- The presentation of this paper can be optimized. I think the description for baseline models can be shrunken, and the authors may expand the introduction of their proposed method in the Methodology section.

- The Experiments part is kind of weak, in terms of evaluation metrics. This paper only considers RMSE. For ERA5 datasets, scientists usually conduct comprehensive evaluations from different perspectives [1-2]. For scientific data (heat diffusion, and fluid dynamics), relative L2 errors are commonly considered. Adding more evaluation metrics can strengthen this paper.

- The writing of this paper is not good enough. There are many typos and grammatical issues. I list several of the issues below. Please do a thorough check.
    - On Page 1, “In contrast, various real-world data is often comes from …”, “is” is redundant.
    - On Page 1, “An example which can benefits significantly…”, “benefits” should be “benefit”.
    - On Page 2, “MSA keep one latent vector per encoded measurement and can access them directly for forecasting.”, “keep” should be “keeps”.

---

**Refs:**

[1] Bi, Kaifeng, et al. "Pangu-weather: A 3d high-resolution model for fast and accurate global weather forecast." arXiv preprint arXiv:2211.02556 (2022).

[2] Pathak, Jaideep, et al. "Fourcastnet: A global data-driven high-resolution weather model using adaptive fourier neural operators." arXiv preprint arXiv:2202.11214 (2022).

**Questions:**

What spatial subsampling methods do the authors use in this paper? It seems not mentioned in the paper.

---

> ### Author Response · Authors · 2023-11-11
>
> Dear Reviewer 7eM6,
>
> Thank you for your review
>
> For the spatial subsampling: For wind nowcasting, we did not apply any subsampling since the partitioning of data into time slices inherently provides this. For other datasets, random subsampling was used.
>
> Your input on the paper's presentation is valuable, and we apologize for the overlooked typos, which will be promptly corrected.
>
> Regarding the evaluation metrics, we actually used a set of 7 comprehensive metrics for the wind nowcasting task, and in the appendix, we benchmarked this task in several different cases: varying forecast periods and examining the impact of context-target distance on performance. Finally, while Table 1 presents MSE values for consistency, the appendix includes comparisons using metrics from related works, including relative L2 errors in some cases. While we agree that having a comprehensive evaluation of the ERA5 results would be nice to have, we think that the MSE results are enough to show that our method is able to be competitive in very different domains.
>
> Considering the strengths you recognized and the fact that the weaknesses you list are mainly presentation-oriented, the 'reject' rating seems disproportionately severe. We are committed to enhancing the paper based on your feedback, so if you have any other concerns, we look forward to possibly addressing them during the discussion period. If this is not the case, we hope you will be open to changing your final rating, especially considering your acknowledgment of our paper's contribution to a less explored yet important domain in scientific ML.

---

> > ### Author Response · Authors · 2023-11-22
> > **End of discussion period**
> >
> > Dear Reviewer,
> > Thanks again for your review. Since we are reaching the end of discussion period, we wanted to ask whether you’ve had the chance to look at our rebuttal. We believe it addresses most of your concerns. If you have any remaining or additional questions, comments, or concerns, it would be great to hear back from you before the period closes. Thanks!

---

### Official Review · Reviewer_nVVV · 2023-10-29

**Soundness:** 2 fair
**Presentation:** 2 fair
**Contribution:** 2 fair
**Rating:** 5
**Confidence:** 3

**Summary:**

In this paper, the authors propose a novel attention mechanism allowing to efficiently apply attention layers to data on non regular grids. The authors demonstrate the effectiveness of the proposed scheme on several applications ranging from PDEs to nowcasting.

**Strengths:**

- The paper is well written;
- The authors demonstrate the soundness of the proposed method on several problems;
- Experiments are convincing;
- In particular, results on the wind nowcasting problem are impressive.

**Weaknesses:**

- The link with the literature is unclear making the novelty of the approach difficult to assess. In particular, works related to motion forecasting (which shows strong similarities with the reference example of this paper) are not referenced [1, 2, 3]. In a context of explosion of attention-based architectures, a more detailed literature review is expected.
- The global presentation of the attention mechanism does not follow the same notations and presentation as is usually the case in the literature and could gained in being clarified.
- The authors alternate between the presentation of a general attention mechanism and a very specific application (wind nowcasting), making it difficult to keep in sight the goal of the paper.

**References**

[1] Nayakanti, Nigamaa, et al. "Wayformer: Motion forecasting via simple & efficient attention networks." 2023 IEEE International Conference on Robotics and Automation (ICRA). IEEE, 2023.
[2] Yuan, Ye, et al. "Agentformer: Agent-aware transformers for socio-temporal multi-agent forecasting." Proceedings of the IEEE/CVF International Conference on Computer Vision. 2021.
[3] Girgis, Roger, et al. "Latent variable sequential set transformers for joint multi-agent motion prediction." arXiv preprint arXiv:2104.00563 (2021).

**Questions:**

I thank the authors for their interesting work.

**Major comments**

1. My first concern is the clarity of the setup. In particular, I found section 3.1 confusing. The apparent will to keep very general notations, while being motivated by a very specific example that not all readers of ICLR papers are familiar with, is misleading. The authors write: "prediction of target values given [...] context and [...] target position. [...] (c_x, t_x) are positions [...] "(c_y, t_y) are measurements." The notations seem to suggest that "c_x" stands for "context at coordinate x" and t_x for "target at coordinate x", but then the network inputs are (c_x, c_y, t_x). An illustration would be very welcome at this stage. Why not move one of the figures in the supplementary at that level? This would clarify the presentation greatly. (Typically, the following sentence : "The context consists of past wind speed measurements, while the targets comprised subsequent wind speed measurements taken at potentially different positions." rather induced more misunderstanding on my side than any clarification. I had to rely on Figure 4 to understand the underlying problematic. Figure 3 is very clear, too.)
2. I find that the notations chosen in equations (1)-(9) is rather uncommon and make it difficult to understand the novelty of the approach. Are the $\varphi, \psi, \phi, \nu$ the usual $\operatorname{softmax}(QK^\top)V$ layers? Integrating (3) and (4) to (5) and (6) would clarify the approach. Similarly, eq. (10)-(13) could be moved to supplementary.
3. Is the proposed method only removing a layer to TFS, as suggested after (9)?
4. It remains unclear to me how the proposed work relates to other works such as [1-5]. Could the authors elaborate on that?
5. "Current state-of-the-art models are graph neural networks and require domain-specific knowledge for proper setup." Graph neural networks are not the only ones employed on spherical data, see e.g. [6, 7]. Could the authors comment on that?
6. Similarly, for irregularly sampled data, Fourier Neural Operators are an option too [8] that have been applied to weather forecasting [9]. Could the authors comment on that?

**Minor comments**

1. I think the abstract could be synthetized in a single paragraph; the last sentence is not necessary.
2. Table 1 is very vague as it is not quantitative (how do the authros evaluate "simplicity"?) and does not adds much to the paper.
1. "as it does not create a bottleneck." --> I think it would be good to define there what the authors mean by bottleneck.
2. "ablations studies" --> "ablation studies"


**References**

[1] Nayakanti, Nigamaa, et al. "Wayformer: Motion forecasting via simple & efficient attention networks." 2023 IEEE International Conference on Robotics and Automation (ICRA). IEEE, 2023.

[2] Yuan, Ye, et al. "Agentformer: Agent-aware transformers for socio-temporal multi-agent forecasting." Proceedings of the IEEE/CVF International Conference on Computer Vision. 2021.

[3] Girgis, Roger, et al. "Latent variable sequential set transformers for joint multi-agent motion prediction." arXiv preprint arXiv:2104.00563 (2021).

[4] Alerskans, Emy, et al. "A transformer neural network for predicting near‐surface temperature." Meteorological Applications 29.5 (2022): e2098.

[5] Bilgin, Onur, et al. "TENT: Tensorized encoder transformer for temperature forecasting." arXiv preprint arXiv:2106.14742 (2021).

[6] Cohen, Taco S., et al. "Spherical cnns." arXiv preprint arXiv:1801.10130 (2018).

[7] Ocampo, Jeremy, Matthew A. Price, and Jason D. McEwen. "Scalable and equivariant spherical CNNs by discrete-continuous (DISCO) convolutions." arXiv preprint arXiv:2209.13603 (2022).

[8] Li, Zongyi, et al. "Fourier neural operator for parametric partial differential equations." arXiv preprint arXiv:2010.08895 (2020).

[9] Pathak, Jaideep, et al. "Fourcastnet: A global data-driven high-resolution weather model using adaptive fourier neural operators." arXiv preprint arXiv:2202.11214 (2022).

---

> ### Author Response · Authors · 2023-11-12
>
> Dear reviewer NVVV,
>
> Many thanks for your thorough review of our paper.
>
> Concerning the weaknesses you mentioned:
>
> 1. We agree that attention-based models seem to be used everywhere. In this work, we show that they possess properties that make them outperform competitors, for instance, the fact that their latent representations are disentangled. This leads to them being a very general type of architecture that can be applied to many problems.
> Regarding the literature on motion forecasting and its link to the reference example of this paper, we believe that while there are some connections—such as the fact that both processes are recorded along trajectories and that in both cases an attention-based model is used—the similarity stops there. Indeed, in our work, we don't rely on the fact that the points are recorded along trajectories; particularly, the goal is not to predict the next trajectory points. Our work mainly focuses on the fact that the data are sparsely recorded in time and space. A set of context points in the case of wind nowcasting contains only a few points per trajectory but many trajectories over the entire space. Hence, the goal here is more to be able to process data from anywhere and to extract a forecast anywhere, rather than being able to complete a trajectory. The rest of the applications, for example, do not have an inherent trajectory structure.
> In more detail, for the specific work you mentioned: Wayformer [1] uses a traditional transformer to condition the prediction of a trajectory given a scene, which is composed of a mix of different types of inputs, but we don't believe this work deals with sparse measurements, and the goal of the model is to decode a trajectory, not to forecast a given phenomenon anywhere in space. Agentformer [2] mixes temporal and social dimensions into a single sequence modeled by a transformer; this seems more focused on forecasting time series with no notion of being able to forecast anywhere in space. Sequential set transformer [3] again is mostly focused on forecasting different futures for agents; in our case, our primary goal is not to forecast the trajectories of planes but to extrapolate reliable wind forecasts from their measurements. However, we are open to adding a sentence in the related work section to distinguish our approach from attention-based models for trajectory predictions.
>
> 2. We are sorry that it was confusing; for the attention part, we tried to stay as close as possible to ViT notation, but additional work was necessary to specify the context and targets, and we think the confusion may have arisen here. More details on this in the answer to comments.
>
> 3. We note your remark; we think our method is general enough to be applied in many cases involving sparse data, that is why we tried to keep our description as general as possible.

---

> ### Author Response · Authors · 2023-11-12
>
> On the major comments:
> 1. We agree that the notation is quite heavy, but we don't think there is a simpler way to describe the problem, as we need to say that the models will process a set of data that can change from sample to sample and that they should be able to generate an output conditional to a set of measurements. We propose to change the sentence you mentioned by "the models are given a set of context points at positions $c_x$ of value $c_y$, and should be able when given target positions $t_x$ to output a corresponding value $t_y$ conditioned on the context." Adding an illustration there would be too much in our opinion, but we propose linking to the one in the appendix.
> 2. Here equations 1-4 are referring to the way of encoding the different quantities $c_x$, $c_y$, and $t_x$ into the models. The problem is that the model processes vectors of the same dimensions without distinction between context measurements (which contain both position and values) and target positions. $\varphi$, $\psi$, $\phi$, and $\nu$ are all multilayer perceptrons.
> 3. No, the proposed model proposes to remove the whole decoder stack and to feed both the encoded input and encoded readout to the same transformer encoder. A traditional transformer contains both an encoder part, which consists of multi-head self-attention with full attention + feedforwards, and a decoder part with causal self-attention and cross-attention with the final layer of the encoder. The final model, therefore, does not have any decoder part and consists only of self-attention with full attention and feedforward layers.
> 4. As transformers are really general architectures, it's not surprising to see them used in a lot of different contexts. In [5], "A transformer neural network for predicting near-surface temperature," it seems that they use a transformer to model a 1D time series, and while it focuses on weather data as well, this work does not focus on how to forecast based on sparse data and doesn't need to encode values at different positions. [6] TENT: Tensorized Encoder Transformer for temperature forecasting seems to use an additional dimension to a transformer-encoder architecture in order to be able to deal with multivariate time series. As far as we understood the paper, each of these time series corresponds to a given city, where we have information at each time step. In our work, this is not the case as measurements are sparse and their positions can vary from example to example.
>
> 5. [6-7] are focused on spherical CNNs, which are a good way to adapt CNNs when dealing with a spherical grid and thus a good candidate for weather forecasting based on a regular spherical grid. We thought that FourcastNet, a graph net approach, was considered state-of-the-art in that regime, but we'll review if there are cases where the state of the art is reached by spherical CNNs and update the related work section if that is the case.
> In our work, however, we consider data that is not sampled along any (spherical) grid. GEN, our considered baseline, uses graph convolutions in its message-passing scheme, which is a way of generalizing from CNNs and spherical CNNs to irregular meshes.
>
> 6. Fourier Neural Operator is a very interesting line of work. Sadly, it relies on having information on regular 2, 3, or 4D grids in order to work, as it uses FFT extensively. This is not the case in our setup. We tried at some point to adapt FNO to our setup by using non-uniform discrete Fourier transforms (nuDFTs), which seemed to be the simplest way of adapting FNO to our case. Sadly, the results were not efficient nor really stable, so we decided not to continue in that direction.
>
> We noted all your minor comments and will update the manuscript accordingly. For table 1, we think it is a nice way of presenting the strengths and weaknesses of the different baselines for a reader who only reads this work diagonally. We found several papers using this kind of notation.
>
> Thank you again for your in-depth analysis of our work. We are at your disposal if any of our answers are not clear enough.

---

> > ### Comment · Reviewer_nVVV · 2023-11-21
> >
> > I thank the authors for their replies. I may be mistaken, but it seems the updated paper has not been attached; could the authors please upload a revised version of the paper in order to better assess their response?

---

> > > ### Author Response · Authors · 2023-11-21
> > >
> > > Dear Reviewer nVVV,
> > >
> > > We have just submitted the revised version of our paper along with the supplementary material. Key updates have been made to the related works section, including rephrasing and the inclusion of an additional section in the supplementary material, as well as in Section 3.1.
> > >
> > > Best regards

---

### Official Review · Reviewer_3Jak · 2023-10-30

**Soundness:** 3 good
**Presentation:** 3 good
**Contribution:** 1 poor
**Rating:** 6
**Confidence:** 3

**Summary:**

This paper tackles making predictions in irregularly sampled (in space and time) sequence modeling tasks.  The net result is a transformer-like architecture that can gracefully scale and adapt to the irregularities.  Some comparisons are then made between the architectures in a synthetic task to understand where the performance gains may be coming from.

**Strengths:**

I think this paper is a nice application paper.  Clearly flexible function approximators are needed for weather forecasting from sparse mobile sensors.  The empirical validation also appears sound, and some of the experiments pulling apart the architectures in Section 5 are a great complement.  The paper itself is also reasonably well written and reads very easily.  Figures are nicely prepared and support the textual exposition.  The supplementary materials and code (although I didn’t actually run it) are incredibly thorough.

**Weaknesses:**

I found this a really difficult paper to review.  The paper is enjoyable to read, fairly self-contained, and the experiments seem to support the hypothesis.  Presenting a simple method _should not_ present a barrier to publication.  This paper is also an absolute A+ in terms of clarity and supporting materials.  Kudos to the authors.

My only real concrete criticism is that I think the technical contribution is minor.  Separating the input encoder is not exactly an awe-inspiring innovation – but it seems to work well on the experiments provided, so I have a hard time pushing back too much on it.  Some of the ablation experiments are nice, but they are not exactly surprising to me – bottleneck layers limit recall necessarily, attention provides clear computation pathways, etc.  But these are still interesting, correct and well-presented investigations.

The opening of the paper discusses a real-world need, but this need is subsequently unaddressed, which makes me think it was just some “window dressing”.  If the authors could propose a concrete application on real-world data, where the predictions from other methods could be verified as being “inadequate” by an understood metric, then I would find that compelling.

I suppose my main lingering concern is simply one of impact.  I don’t believe the experiments here will convince many practitioners to adopt this method, or that the insight required for this method is dramatically different to simple “a transformer”.  The main “deep” baselines have all also been proposed by the authors, and so it is difficult to know if the MSA method is better, or if the baselines have just been poorly tuned.  Comparing to published literature removes this uncertainty.  However, again, I concede that I do not know what experiments _would_ convince me of this, or to what papers I would look to benchmark against.  Maybe comparison to examples in [1-3] would be compelling?

Maybe comparatively then, I think the strength of the technical contribution, range and depth of the experimental evaluation, and general impact of this paper are lower than other papers I have recommended be accepted.  While this does not mean I recommend the paper be rejected, I find it difficult to ardently campaign for its inclusion.

I invite the authors to push back against observations, particularly highlighting the innovations over existing models in the field or the general applicability of the solution

[1]  Learning Temporal Evolution of Spatial Dependence, Lan, PMLR, 2022.

[2]  NEURAL SPATIO-TEMPORAL POINT PROCESSES, Chen+, ICLR, 2021.

[3] A spatio-temporal statistical model to analyze COVID-19 spread in the USA, Rawat+, J Appl Stat, 2021.

**Questions:**

*Q.1.*:  Can the authors clarify what the “transformer-decoder” and “transformer-encoder” architectures are.

*Q.2.*:  Why do the “transformer-encoder” in (5) and (7) have different input and return types?

---

> ### Author Response · Authors · 2023-11-12
>
> Dear Reviewer 3Jak,
>
> First, many thanks for your kudos and for your overall nice review.
>
> We'll answer your questions first. Q1: We thought that these notions were well-defined. The transformer encoder consists of a series of blocks consisting of self-attention mechanisms, which utilize full attention, and feed-forward networks, interleaved with residual connections and layer norms. A transformer decoder includes an additional cross-attention layer. In the decoder, the self-attention layers traditionally use causal masking, but we removed that in our implementation as there is no problem if the target positions see each other, and we use the architecture to predict all measurements in one pass, not autoregressively. For reference, consider the official PyTorch implementation or the original encoder stack from "Attention is All You Need." Q2: We noted it that way to state that in (5), the transformer encoder takes as input the two sets of encoded contexts and encoded target positions, which in practice are concatenated. Since the output is of the same dimension, we have corresponding processed latents for each context point and each target position, but only the output corresponding to the target position is used later on. For (7), we only pass the context measurements to the transformer encoder. We tried using concatenation notation for (5), but it was heavier, so we stuck to a more general mathematical formulation.
>
> For your remark concerning the real-world need, we wanted to state that this work was developed with an industrial partner with a production application for the wind nowcasting task in the context of air traffic control in mind. This is also the reason why we focused on an architecture that is robust and reliable. And this is why we preferred offering clear justifications for the architecture instead of overcomplicating it. In this context, the CNP baseline was outperformed by a previously considered baseline. Adapting GENs for this task required more design choices, for example, how to define the underlying graph given the considered space, how to choose the links between nodes, and a careful hyperparameter exploration regarding the learning rate for the position of the graph. While we can't cite a specific example where this method underperforms compared to MSA, our model consistently offered better performance while being easier to implement.
>
> Concerning the paper you mention:
>
> [1] "Learning Temporal Evolution of Spatial Dependence with Generalized Spatiotemporal Gaussian Process Models" seems to use a specific Gaussian process to model spatiotemporal models. While Gaussian processes have strong theoretical properties, they can be hard to implement, require very specific knowledge from the final user, and are hard to scale to a large number of inputs. We considered Conditional Neural Processes (CNP) as a baseline, which were introduced to solve these problems.
>
> [2] "Neural Spatio-Temporal Point Processes" appears to model discrete events and their effects that can happen irregularly in time and space and are more related to density estimation of time series data than conditioned prediction.
>
> [3] "A Spatio-Temporal Statistical Model to Analyze COVID-19 Spread in the USA" seems to be similar to [1] in that they rely on a separable GP. We agree that having additional Neural ODE and GP baselines, or evaluating our method against them on their benchmark would be nice to have, but we thought that our benchmark was comprehensive enough to prove our point.
>
> Considering the general application, we think that our method can be useful anytime there is a need to manage a fleet of moving vessels, for example in the case of airspace, but other applications could be drone coordination or maritime fleet management. We agree that, in essence, other practitioners could probably have the idea of using transformers for that task by themselves, but even in that case, our work would provide a way of adapting the general architecture for that specific task, a proof that it works, and some justifications about why it is a good idea to use transformers in that case, along with comparisons with existing baselines.
>
> We thank you again for your clear and valuable opinion and your time reviewing our paper.

---

> > ### Comment · Reviewer_3Jak · 2023-11-19
> > **Sticking**
> >
> > To the authors,
> >
> > Thank you for your response and for clarifying my two questions.  There are two outstanding ``issues'' as I see it:  real-world impact, and technical contribution.  Unfortunately, I am not particularly swayed by the authors response to either of these.  The papers I listed are in slightly different domains, agreed.  However the authors should look for adjacent fields/applications to demonstrate and compare their method on, and maybe these applications are *close enough*.  This will help increase the reach and impact of the paper, and help cement the utility of the method.  I appreciate the authors candor r.e. technical contribution, but I think demonstrating that a tweak to flexible architecture works well in a limited application domain is on the lower end of the contribution spectrum.
> >
> > To make my recommendations concrete and actionable (and hopefully useful), I suggest the authors find comparable domains to apply and benchmark their method on.  They may have industrial partners interested in weather nowcasting, but it is probably just too niche of an application to get mass-market appeal and traction at top conference venues.  Continuing to develop the methodology on these applications will hopefully reveal technical innovation that will help bolster the contribution.
> >
> > This a great start, but I personally don't feel it is quite at the bar for publication.
> >
> > Good luck,
> >
> > 3Jak

---

> > > ### Author Response · Authors · 2023-11-21
> > >
> > > Dear Reviewer,
> > >
> > > Thank you for your thorough review and constructive feedback. We truly appreciate that you took considerable time and effort to analyze our work and identify areas for improvement. We regret that our previous response did not convince you, and we apologize if it came across as overly defensive.
> > >
> > > We are grateful that initially, you found no compelling reason for rejection and acknowledged the value of our study within its current scope. Your insights on extending the scope and contributions of our paper are well-received, and we view them as valuable directions for future research. While our intention was to simplify the model, focusing on the essentials for functionality, we understand that this approach might seem limiting and potentially under-contributing in some aspects. However, we firmly believe there is significant merit in pursuing simplicity and applicability.
> > >
> > > Thank you again for your time and the effort you've dedicated to providing us with such detailed and helpful feedback.

---

### Official Review · Reviewer_jjJ1 · 2023-10-31

**Soundness:** 2 fair
**Presentation:** 2 fair
**Contribution:** 2 fair
**Rating:** 3
**Confidence:** 3

**Summary:**

A modularized transformer is proposed, where context and target information is encoded in separate modules to process spatially irregularly distributed data. In a comparison against a graph neural network and conditional neural processes, the proposed method shows improved performance on a wide variety of datasets.

**Strengths:**

_Originality:_ Neither the task nor the method seem to be particularly new. The modularization of the transformer into different compontents is appealing, though.

_Quality:_ Given the conducted experiments, it is not all clear whether the proposed model holds what the authors promise and whether it is state-of-the-art. My biggest concern is the limited amount of related work cited in the manuscript and I do not see the field of related work well explored. There are numerous works that operate on irregular meshes, e.g., [[1]](http://arxiv.org/abs/2302.10803), [[2]](http://arxiv.org/abs/2207.05209), [[3]](http://arxiv.org/abs/2210.05495), [[4]](http://arxiv.org/abs/2210.00612), [[5]](http://arxiv.org/abs/2103.01342), which I would require to see in a comparison study.

_Clarity:_ I had great difficulties to understand large parts of the manuscript. In particular, the data description was unclear to me and the purpose of some experiments remained vague to me (see questions below)

_Significance:_ The significance of this method and its contribution to the field are unclear to me. The results on the wind forecasting, in particular, are promising but need more contextualization with other models accomplishing similar tasks.

_Further comments_:
- The manuscript comes with a very detailed appendix and supplementary material, which is a great resource for exploration! Adding the Navier-Stokes figure that shows context and target points to the main document would be helpful, for example.
- The number of benchmark datasets is great; ideally, this should be complemented with more competitive models form the litareture.

**Weaknesses:**

1. Several unclear details. What is the spatial and temporal distance of measurements in the different datasets? what do you mean with bottleneck? Do you refer to information compression when projecting to latent space (I could not find where it is defined)?
2. The naming of context and target was confusing to me. In Section 3.1 you specify: `Data is in the form of pairs of vectors $(c_x,c_y)$ and $(t_x,t_y)$ where $c_x$ and $t_x$ are the position and $c_y$ and $t_y$ are the measurements (or values).` It is unclear to me whether $x$ and $y$ are referred to inputs and targets or if $c$ and $t$ denote inputs (context) and targets. Maybe use $p_x, p_y$ for positions of input and target, respectively, and $v_x, v_y$ for the vectors holding the values for inputs and targets, respectively.
3. I am not sure whether the traditional transformer with positional encoding has been benchmarked here. Neither GEN nor CNP seem to resemble the traditional transformer. The vanilla transformer would constitute an indispensible baseline.
4. The context information retrieval experiment was difficult to grasp and I could not quite understand why other models (with a bottleneck) fail. Isn't the problem just a matter of spatial sampling? That is, when having sufficient point measurements, wouldn't all model perform well? Alternatively, wouln't a larger latent vector solve the issue as well?
5. I do not quite understand your statement in Section 5.1 `The higher $f$ is, the more difficult the function becomes, as local information becomes less informative about the output.` To my understanding, larger values of $f$ increase the frequency, making local information more important than far distant information.
6. Unclear what runtime and/or memory consumption the introduced method induces. According to Figure 5, MSA seems to employ a 8x8 grid, which might be expensive to handle computationally. Some information about computation, memory, and accuracy trade-offs would be helpful to justify the method. How big is the context size used in your experiments and when does the model break down (du to scaling, as mentioned in the last paragraph of the conclusion)?
7. Alhtough I do like the analysis, Figure 6 seems trivial to me. Doesn't the number of required gradient steps relate directly to the receptive field of the model and, hence, to the number of message passing steps required to propagate information through the grid?

**Questions:**

1. Given that MSA foregoes with any position information, how can it be able to predict wind direction and speed from surrounding sources? Wouldn't it be essential to know which winds are observed where in space?
2. What does the color code in Figure 4 resemble, is it an error?
3. Similarly, what does the x-axis in Figure 5 show, are these neurons in the output layer?
4. How do the benchmarked methods perform with different numbers of context points?

---

> ### Author Response · Authors · 2023-11-11
>
> Dear reviewer jjj1,
>
> Thank you for your time reviewing our paper. We will try to address your concerns regarding related works, the weaknesses you highlighted, and your questions.
>
> First, on the related works:
> The five papers you are citing focus on mesh-based learned physics simulators. We believe that the setup studied in our paper is not the same, and these methods do not apply here; that's why we didn't consider them as baselines.
>
> [1] Eagle: Large-Scale Learning of Turbulent Fluid Dynamics with Mesh Transformers
> The data inputs are meshes that are generated by numerical solvers (Ansys Fluent), which means that even if the graph is irregular, the models always have values at the nodes of the mesh, which is consistent across all timesteps of all different episodes.
>
> We are trying to extrapolate sets of measurements that can be measured anywhere, where the number of measurements and the positions of measurements can vary from set to set and are not the same between context and targets. That is why using a mesh is not straightforward; actually, if we wanted to do so, we would need a spatial interpolation scheme for converting the measurements to the mesh, and then a second one to go from the mesh to the targets. That's actually what GEN (the considered baseline) does. If we wanted to adapt EAGLE to our case, we would need to add interpolation after the encoder and after the decoder, and so the final method would ultimately be a variant of GEN with some additional clustering, graph pooling, and attention. This final model would suffer from the same problems that GEN has. Note as well that we considered some message-passing scheme with attention in our Appendix 8, and it didn't help.
>
> [2] Fourier Neural Operator with Learned Deformations for PDEs on General Geometries
> Introduce geo-FNO, which learns a mapping from an irregular mesh to a regular one on which FNO (based on FFT) can be applied, and finally maps from the regular lattice back to the irregular mesh. While this approach is interesting, it still needs to be applied to dense meshes. In our case, the space is sparsely populated with measurements: in a given set of data, we only have measurements in some parts of the space, not everywhere but irregularly.
> We tried at some point to adapt FNO to our setup by using nonuniform discrete Fourier transforms (nuDFTs), which seemed to be the simplest way of adapting FNO to our case. Sadly, the results were not efficient nor really stable, so we decided not to continue in that direction.
>
> [3] MAgNet: Mesh Agnostic Neural PDE Solver
> This work uses the same encode-interpolate-forecast structure as Graph Element Networks and only differs because they use data from a parent mesh (so they don’t need to interpolate from the measurement to the mesh) and that it is applied to time series. Again, if we wanted to adapt their work as a baseline, we would probably end up with something very similar to Graph Element Network.
>
> [4] Multiscale Meshgraphnets
> Is an extension of Meshgraphnet which falls back to a variant of GEN if we tried to adapt it to our sparse measurements setup. Actually, we mention Meshgraphnets in the related works, second paragraph. But we agree we could make that point clearer. We propose adding the following sentence: "[...] irregular but fixed mesh. While these approaches are achieving impressive performances in their domains, using a mesh-based approach has some drawbacks when working with sparse measurements as we need a way of mapping the measurement to the mesh and most of its nodes would have no related measurements."
>
>
> [5] Reinforcement Learning for Adaptive Mesh Refinement
> This seems to use reinforcement learning to define the proper mesh for finite element methods. Again, having a fixed irregular mesh would be nice, but it does not help as, in our case, a set of data is sparse and does not "cover" the entire space, so most of the nodes of such a mesh would be empty most of the time.
>
> So, to summarize, in our case, we cannot rely on meshes because the data is sparse and varies from set to set. As a practical example, where our work is used in practice, consider an airspace. We can have access only to measurements where planes are, and most of the space at a given time is empty. The only way of using a graph neural network, since the measurements can come from anywhere, would be to interpolate them onto the graph, then process them, and then extrapolate to where we have the queries. And this is exactly what GEN does. The other baseline we consider, CNP, uses a different approach but can also take a set of sparse measurements as input and interpolate from them as well, which these mesh-based architectures cannot do.

---

> > ### Author Response · Authors · 2023-11-11
> >
> > on the weaknesses :
> > 1. On the spatial and temporal distance: Good point, we will add the corresponding order of magnitude in the appendix. By "bottleneck," we mean that different context measurements can be mapped to the same latent vector, which then creates some problems. In some pathological cases, when we gave these models some measurements as input, they were not able to restore them all. We think the first part of Section 5 was clear, but we will improve the phrasing of that part.
> > 2. We note your remark and will try to make the notation less confusing.
> > 3. We actually adapted a traditional transformer (TFS) in that case as a baseline. We showed that it did not suffer from the bottleneck problem and offered some performance gains while being already a quite simple and general architecture. But we showed that we don't need an encoder-decoder architecture and using only a transformer encoder leads to even better performances and is even simpler.
> > Using positional encoding for the position in the sequence in that case would be a bad idea as it breaks one inductive bias: the architecture should be invariant to the order of context measurements, which is why we removed them.
> > 4. We apologize if it was hard to comprehend. All these models are supposed to extract data or forecasts from a context set. A property that we expect them to have is that when given a set of N measurements, they should be able, when queried properly, to retrieve any one of these measurements. As another example: if we think about a general interpolation formula between N measurements, we want to guarantee that the resulting interpolation matches the measurement points at the measurement positions. This is the property that we want the different models to have, and we showed that models with a bottleneck (GEN, CNP) do not have this property in some cases.
> > 5.  `larger values of increase the frequency, making local information more important than far distant information.` Exactly. If two positions are close and end up being mapped to the same latent vector, when the space is smooth it does not matter as the measurements probably have the same value. When we increase the frequency, it is not the case anymore, and that’s when it becomes problematic for the model: if two measurements of different values are mapped to the same latent vector, and the model should use that vector to retrieve the value, in practice that's when it fails.
> > 6. In Figure 5, the grids correspond to the GEN graphs, compared to MSA and TFS. The attention mechanism scales quadratically with the context and target size. For GEN, it depends on the message-passing scheme, but it is at least of the order of (number of nodes) x (number of steps), which can grow quickly if we need a large graph to encode the sparse measurements, while CNP is linear. We use a context size containing up to a thousand context points for a given sample in the training set. In this regime, as mentioned in the conclusion, all the methods are working and the scale is not a problem.
> > 7. Perhaps there is confusion here as the grids refer to GEN. What we aimed to show is that not having a bottleneck helps during learning.
> >
> > On the questions:
> > 1. We forego the positional encoding for the position in the sequence; each input consists of both the geographical positions of the measurements and their values, both encoded.
> > 2. Exactly, showing when models with a bottleneck fail.
> > 3. No, these are the latent vectors of the corresponding models, so last layer of the encoder for both MSA and TFS, and the latent nodes after message passing but before decoding for GEN.
> > 4. In the wind nowcasting experiment, the number of measurements per set varies from less than 10 up to 1000, and MSA obtains better performance overall.
> >
> > We hope this response has clarified some of your questions, particularly concerning why mesh-based approaches are not suitable candidates for baselines in this case. Please let us know if anything remains unclear, as we would be happy to provide further information if necessary.

---

> > > ### Author Response · Authors · 2023-11-22
> > > **End of Discussion Period**
> > >
> > > Dear Reviewer,
> > > Thanks again for your review. Since we are reaching the end of discussion period, we wanted to ask whether you’ve had the chance to look at our rebuttal. We believe it addresses most of your concerns. If you have any remaining or additional questions, comments, or concerns, it would be great to hear back from you before the period closes. Thanks!